# Sulfenamide-enabled *ortho* thiolation of aryl iodides via palladium/norbornene cooperative catalysis

Renhe Li [1], Yun Zhou[1], Ki-Young Yoon[1], Zhe Dong [2] & Guangbin Dong [1]

Poly-substituted aromatic sulfur compounds are widely found in pharmaceuticals, agrochemicals and organic materials. However, the position that a sulfur moiety can be introduced to is largely restricted to a pre-functionalized site; otherwise, use of electronically biased substrates or auxiliary groups that direct catalysis is required. Here we report a general *ortho* thiolation of common aryl and heteroaryl iodides via palladium-norbornene cooperative catalysis. Using this approach, an aryl or alky sulfur moiety can be site-selectively introduced at the arene *ortho* position without using sterically or electronically biased substrates. The arene *ipso* functionalization is simultaneously achieved through Heck, Suzuki or Sonogashira termination. The reaction is enabled by a unique class of electrophiles in palladium-norbornene cooperative catalysis, which are sulfenamides derived from seven-membered lactams. The broad substrates scope and high chemoselectivity could make this method attractive for synthesis of complex sulfur-containing aromatic compounds.

---

[1] Department of Chemistry, University of Chicago, Chicago, IL 60637, USA. [2] Merck Center for Catalysis at Princeton University, Princeton, NJ 08544, USA. Correspondence and requests for materials should be addressed to Z.D. (email: wdzdao@gmail.com) or to G.D. (email: gbdong@uchicago.edu)

Aromatic sulfur compounds are commonly found in drugs[1], agrochemicals[2], organic electronics[3], and polymers[4] (Fig. 1a). In addition, aryl sulfides often serve as versatile intermediates to access the corresponding sulfoxides[5], sulfones[6], and benzothiophenes[7]. Common ways to prepare aryl sulfides heavily rely on nucleophilic aromatic substitution[8] and cross-coupling reactions[9] between aryl halides and thiols. Both methods form carbon−sulfur (C–S) bonds at the *ipso* position of aryl halides; thus, the position of the installed sulfur moiety is restricted by the position of the halide. On the other hand, C–H thiolation offers an attractive approach to introduce sulfur to an non-pre-functionalized position;[10] however, control of site-selectivity generally requires use of directing groups[11,12] or electron-rich arenes[13]. Hence, a general method that site-selectively introduces sulfur functional groups to unbiased and unactivated arene positions would be highly attractive for preparing multi-substituted aromatic sulfur compounds. This has motivated us to explore the approach using palladium/norbornene (Pd/NBE) cooperative catalysis.

Pd/NBE cooperative catalysis[14–16], originally discovered by Catellani[17], has emerged as a useful tool for preparing multi-substituted arenes[18–33]. Compared to the conventional arene functionalization, this approach enables simultaneous functionalization of arene vicinal positions regioselectively using simple aryl halides as substrates (Fig. 1b). Specifically, through forming an aryl-NBE-palladacycle (ANP) intermediate, a nucleophile and an electrophile are coupled at the arene *ipso* and *ortho* positions, respectively. While the scope of nucleophiles in this reaction is broad[14–16], finding suitable electrophiles that can participate in the Catellani reaction nevertheless remains a formidable challenge[34,35], because the electrophile must react with ANP selectively in the presence of Pd(0) species and has to be compatible with the nucleophile and NBE. Currently, the scope of electrophiles is mainly limited to carbon[17,36–41] and nitrogen[42]-based reagents. In 2017, Yu used aryl chlorosulfates for a directed Pd(II)-catalyzed *meta* chlorination of arenes;[43] however, compatibility of this reagent with Pd(0) catalysts could be a concern[44].

More recently, Zhang[45] and Cheng[46] independently reported an interesting *ortho* silation with disilanes; unfortunately, NBE cannot be extruded in this reaction. Clearly, it would be attractive if other elements, besides C and N, could be introduced at the arene *ortho* position in the Pd(0)-catalyzed Catellani reaction. Herein, we report a Pd/NBE-catalyzed *ortho* thiolation of aryl iodides, which is enabled by sulfenamide-type electrophiles (Fig. 1c). This approach provides a general platform to introduce various sulfur moieties to the arene *ortho* positon and simultaneously install other functional groups at the arene *ipso* position. The generality, scability and high chemoselectivity could make this method attractive for preparing complex sulfur-containing aromatic compounds.

## Results

**Hypothesis.** Compared to other *ortho* functionalizations, *ortho* thiolation exhibits its unique challenges. First, many electrophilic sulfur-based compounds, such as PhSSPh or PhSCl, readily react with Pd(0)[47], therefore preventing arene functionalization. Second, thiolates (RS−) are known as strong ligands for soft Pd species; thus, decomposition of the thiolation agent would likely generate RS− that could lead to direct *ipso* thiolation[48]. Hence, developing a stable but also reactive electrophilic thiolation agent would be a key for realizing the *ortho* thiolation reaction. Based on our prior efforts on developing the *ortho* amination reaction[42], sulfenamides[49,50] were anticipated to be a suitable electrophile for the Pd/NBE catalysis for two reasons (Fig. 1d): (1) the electronegativity ($E_{neg}$, Pauling scale) difference between N and S matches well with that between O and N;[51] (2) analogous to the *ortho* amination, the amide carbonyl could serve as a directing moiety to facilitate selective reactions with ANP. Thus, we hypothesized that sulfenamides might show similar stability and reactivity as *O*-benzoyl hydroxylamines. It is noteworthy that, during the review process of this work, an interesting *ortho* thiolation using thiosulfonate reagents was reported by Gu[52].

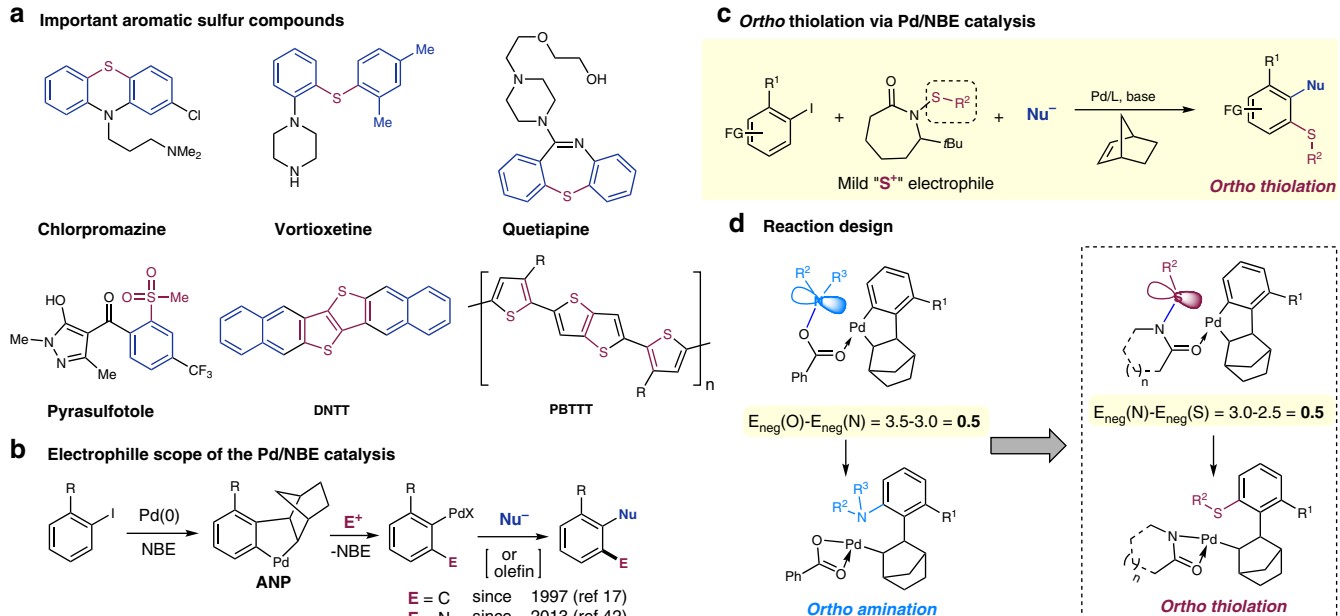

**Fig. 1** Project background and design for *ortho* C–H thiolation via Pd/NBE cooperative catalysis. **a** Important drugs, agrochemicals, organic electronics, and polymers that contain aryl-sulfur bonds. **b** Established *ortho* functionalization of aryl iodides with different electrophiles via Pd/NBE cooperative catalysis. **c** This work describes an *ortho* thiolation using sulfenamides as the electrophilic thiolation reagent. **d** Design of *ortho* C–H thiolation reagents. E electrophile, Nu nucleophile, $E_{neg}$ electronegativity

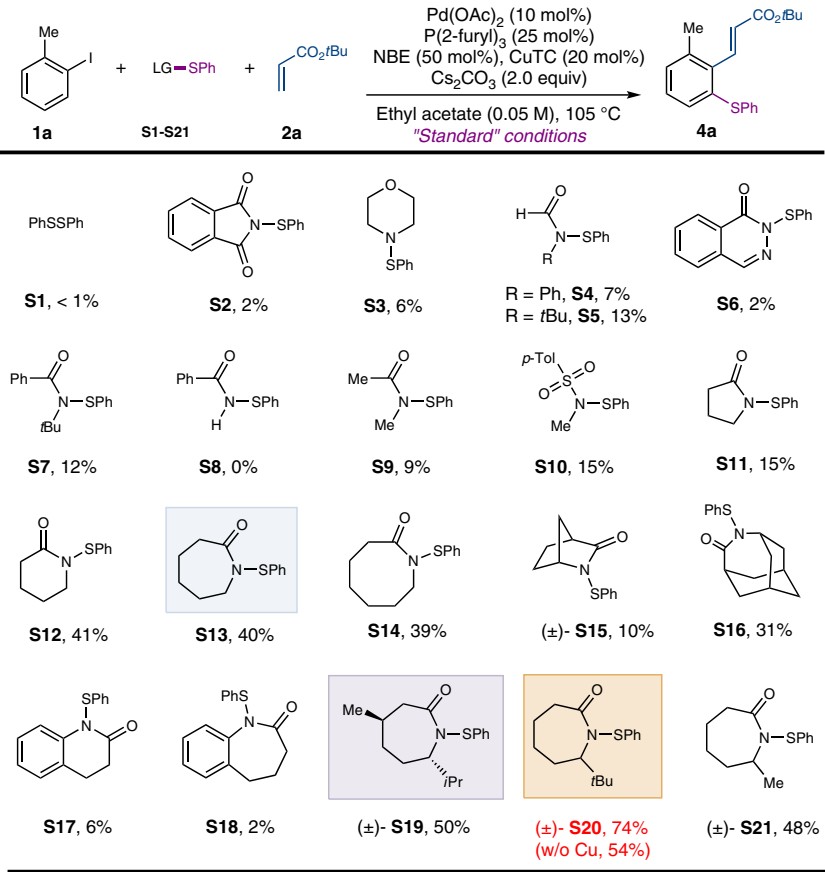

**Fig. 2** Electrophiles for the Pd/NBE-catalyzed *ortho* thiolation of aryl iodides. The reaction was run with **1** (0.15 mmol), **2** (0.30 mmol), sulfur electrophile (0.30 mmol), Pd(OAc)$_2$ (0.015 mmol), P(2-furyl)$_3$ (0.0375 mmol), NBE (0.075 mmol), Cs$_2$CO$_3$ (0.30 mmol), and CuTC (0.03 mmol) in ethyl acetate (3.0 mL) at 105 °C for 12 h. The yield was determined by $^1$H-NMR using 1,3,5-trimethoxylbenzene as the internal standard. LG leaving group, CuTC copper (I) thiophene-2-carboxylate

**Fig. 3** X-ray structures of selected lactam-derived sulfenamides. The nitrogen—sulfur bond lengths are labeled in ångström (Å)

**Optimization of reaction conditions**. To test this hypothesis, a range of sulfenamide-based thiolation agents were examined with 2-iodotoluene (**1a**) as the standard substrate, and the *ipso* position was functionalized via Heck termination with acrylate **2a** (Fig. 2). As a control experiment, PhSSPh **S1**, previously used in the Pd-catalyzed C–H thiolation[11,12], gave almost no desired product with a low conversion of **1a**. In contrast, various sulfenamides indeed afforded the desired *ortho* thiolation product (**4a**). First, neither imide-derived or amine-derived sulfenamides (**S2** and **S3**) were as effective

as amide-based ones. In particular, the lactam-derived sulfenamides (**S11–S21**) were found most reactive. Interestingly, the six, seven, and eight-membered sulfenamides (**S12–S14**) gave significantly improved yields compared to the five-membered one (**S11**). Use of more strained or benzofused lactams (**S15–S18**) gave inferior results. Surprisingly, increasing the bulkiness around the lactam nitrogen with an adjacent isopropyl group significantly enhanced the yield (**S19**). Ultimately, the optimal result was obtained using the tert-butyl-substituted sulfenamide **S20**.

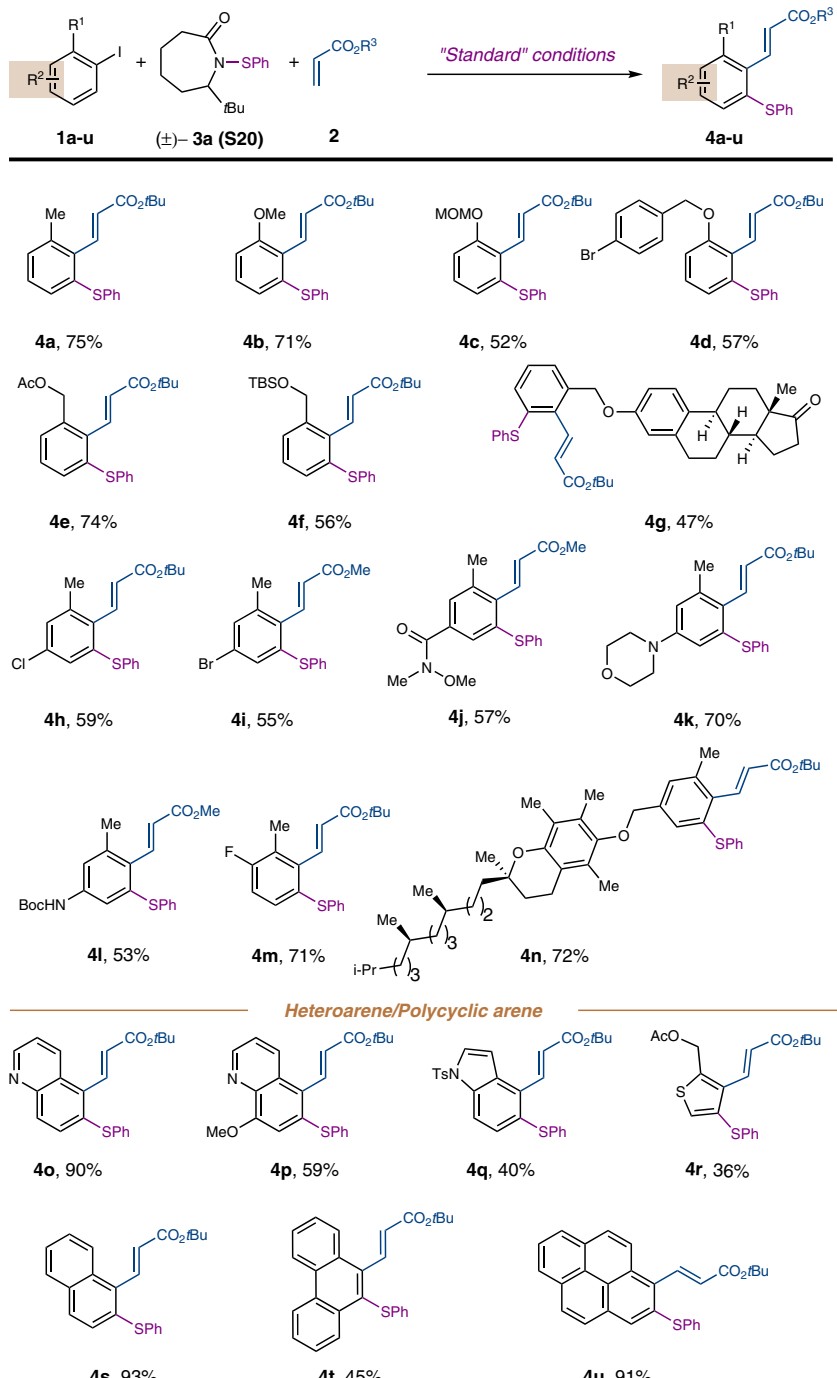

**Fig. 4** The aryl iodide scope of the *ortho* thiolation. All reactions were run with 0.2 mmol **1**, 0.4 mmol **3a** (**S20**), and 0.4 mmol **2** in ethyl acetate (0.5 M) at 105 °C for 12 h. Isolated yields are reported. For detailed experimental procedure, see Supplementary Fig. 4

To understand these counterintuitive results in terms of the role of the bulk substituent, X-ray crystal structures of **S11–S14**, **S19** and **S20** analog (**3c**) were obtained (Fig. 3 and Supplementary Figs. 10–16). A clear trend is that increasing the steric hindrance around the amide moiety elongated the N–S bond, which correlates to the performance of these reagents. Hence, the tert-butyl group in **S20** weakened the N–S bond, thereby making it more reactive. Note that adding copper(I) thiophene-2-carboxylate (20 mol%) enhanced the yield, which may serve as a thiolate scavenger (for full control experiments, see Supplementary Table 1).

**Substrates scope of aryl iodide**. With the optimized conditions in hands, the aryl iodide scope was examined first (Fig. 4). Different substituents at the *ortho* position of aryl iodides were tolerated, including methyl (**4a**), methoxy (**4b**), MOM ether (**4c**), 4-bromobenzyl ether (**4d**), acetate and silyl-protected benzyl alcohols (**4e** and **4f**), and an estrone derivative (**4g**). In addition, a broad range of functional groups were compatible, such as aryl chloride (**4h**), aryl bromide (**4i**), Weinreb amide (**4j**), dialkyl aniline (**4k**), carbamate (**4l**), fluoride (**4m**), and Vitamin E moiety (**4n**). Importantly, the reaction is suitable for a variety of heteroarenes and polycyclic arenes, including quinoline derivative

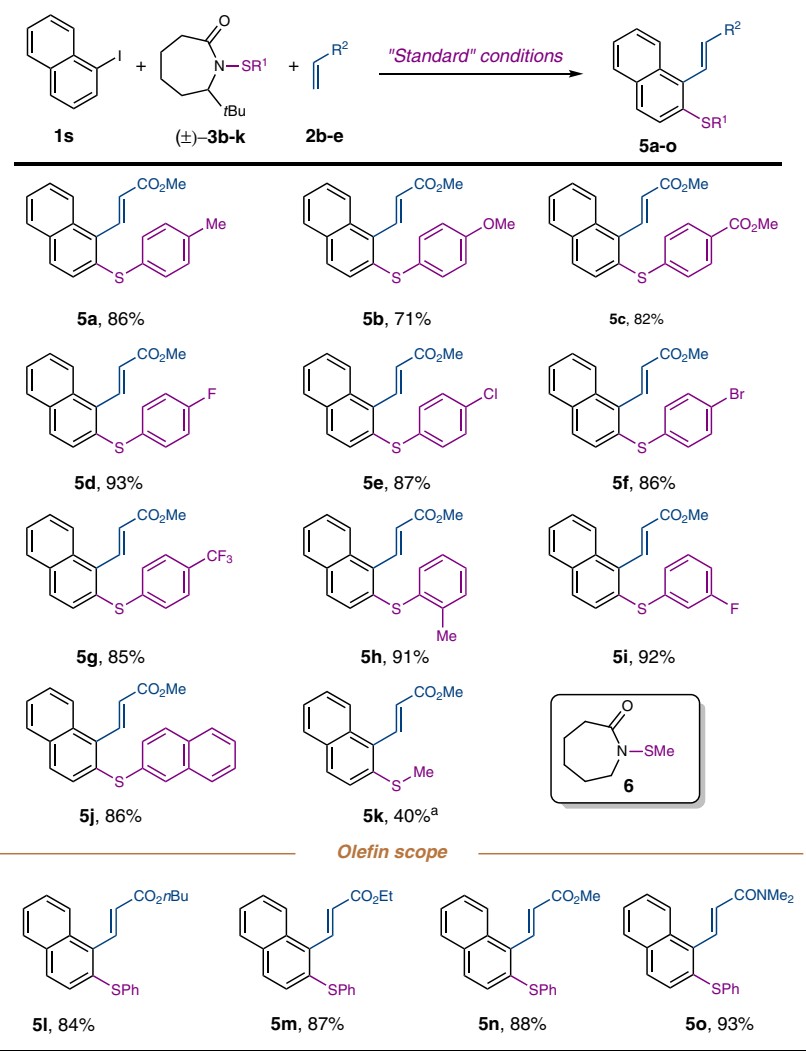

**Fig. 5** The sulfenamide and olefin scope of the *ortho* thiolation. All reactions were run with 0.2 mmol **1s**, 0.4 mmol **3** and 0.4 mmol **2** in ethyl acetate (0.5 M) at 105 °C for 12 h. Isolated yields are reported. [a] Sulfenamide **6** was used instead of **3**. For detailed experimental procedure, see Supplementary Fig. 4

(**4o** and **4p**), indole (**4q**), thiophene (**4r**), naphthalene (**4s**), phenanthrene (**4t**), and pyrene (**4u**).

**Substrates scope of thiolation agents**. Next, the scope of the thiolation agents and the olefin coupling partners was explored (Fig. 5). Besides PhS–, other aryl sulfur groups that contain electron-donating or withdrawing groups could be introduced at the *ortho* position in good to excellent yields. Notably, the *ortho*-substituted aryl sulfide (**5h**) still afforded a high yield of product. While it was challenging to prepare the corresponding alkyl thiolation agents based on the *t*Bu-substituted lactam, use of simple ε-lactam-derived sulfenamide **6** nevertheless delivered the desired methylthiolated product (**5k**) in moderate efficiency. In addition to *t*Bu acrylate, other acrylates and acrylamides (**5l–5o**) were also competent coupling partners for *ipso* functionalization.

**Synthetic application**. From a practical viewpoint, the lactam byproduct **3a'** was recovered in 86% yield after the reaction, which could be used to regenerate the sulfenamide reagent (Fig. 6a). The reaction is scalable: a high yield was still obtained on a gram scale (Fig. 6b). Besides aryl sulfides, the corresponding sulfoxides and sulfones could be conveniently accessed through selective oxidation of the *ortho* thiolation product (Fig. 6c). In addition to Heck coupling, preliminary success has been obtained with Suzuki quench

(Fig. 6d) and Sonogashira quench (Fig. 6e) to install an aryl group or alkyne group at the *ipso* position, respectively[14–16].

## Discussion

In summary, a unique class of electrophilic thiolation reagents, sulfenamides, is developed for the Pd/NBE catalysis, which enables *ortho* thiolation of a wide range of aryl and heteroaryl iodides. The broad substrate scope, scalability, and high chemoselectivity could make this method attractive for complex molecule synthesis. The substituent effect observed in tuning the sulfenamide reactivity could have implications beyond this work. Efforts on expanding the reaction scope and understanding the detailed mechanism of the C–S bond formation are underway.

## Methods

**General procedure of the Pd/NBE-catalyzed *ortho* thiolation**. To a flame-dried 7.0 mL vial (vial A) was added palladium acetate (4.6 mg, 0.02 mmol, 10 mol%), copper(I) thiophene-2-carboxylate (7.6 mg, 0.04 mmol, 20 mol%), tri(2-furyl) phosphine (11.6 mg, 0.05 mmol, 25 mol%), and aryl iodide (0.2 mmol, 1.0 equiv). The thiolation agent (0.6 mmol) was added to another 4.0 mL vial (vial B). These two vials were then transferred into a nitrogen-filled glovebox without caps. In glovebox, cesium carbonate (130.4 mg, 0.4 mmol, 2.0 equiv) was added to vial A before a solution of norbornene in dry ethyl acetate (0.5 mL, 0.1 mmol) was transferred to the same vial. To the 4.0 mL vial B containing thiolation agent was added 0.75 mL dry ethyl acetate, and then two thirds of

**Fig. 6** Synthetic applications. **a** Recovery of lactam **3a'**. **b** Gram scale reaction. **c** Selective oxidation of an aryl sulfide to a sulfoxide and a sulfone. **d** *Ipso* functionalization via Suzuki coupling. **e** *Ipso* functionalization via Sonogashira coupling

this solution (0.5 mL, 0.4 mmol, 2.0 equiv) was transferred into vial A, before another 3.0 mL dry ethyl acetate and acrylate **2** (0.4 mmol, 2.0 equiv) were added. Vial A was then tightly sealed, transferred out of glovebox and stirred on a pie-block preheated to 105 °C for 12 h. After completion of the reaction, the mixture was filtered through a thin pad of silica gel. The filter cake was washed with ethyl acetate and the combined filtrate was concentrated under vacuum. The residue was purified via silica gel chromatography to yield the desired *ortho* thiolation product.

## Data availability

Experimental procedures (Supplementary Figs. 1–9) and characterization data (Supplementary Figs. 10–149) are available within this article and Supplementary Information. CCDC: 1906766 (S11), 1906767 (S12), 1906768 (S13), 1906770 (S14), 1906771 (S19), 1906769 (**3c**), and 1906772 (**4e**) contain the supplementary crystallographic data for this paper. These data can be obtained free of charge from the Cambridge Crystallographic Data Center via www.ccdc.cam.ac.uk/data_request/cif.

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

## Acknowledgements
Financial supports from the University of Chicago and NIGMS (1R01GM124414–01A1) are acknowledged. Y.Z. thanks CSC for a fellowship. Mr. Jianchun Wang is acknowledged for helpful discussions.

## Author contributions
Z.D. discovered the transformation and performed the preliminary optimization. R.L. and Y.Z. performed the optimization, subsrate scope and synthetic applications. K.Y. performed the X-ray crystallography study. G.D. guided the project. Z.D. and G.D. conceived the idea and wrote the paper with input from all authors. All authors analyzed the results and commented on the paper.

## Additional information

**Competing interests:** The authors declare no competing interests.

