## [Peer Review File · Nature Communications]

Reviewers' comments:

Reviewer #1 (Remarks to the Author):

This manuscript by Dong describes an interesting sulfenamide-enabled ortho-thiolation of aryl iodides via Pd/norbornene cooperative catalysis. Inspired by ortho-amination reagents, they designed ortho-thiolation reagents by considering the coordination between carbonyl and palladium, and the electronegativity of sulfur and nitrogen. It is a pioneering idea. Meanwhile, X-ray crystal structures of S11-S14, S19 and S20 clearly demonstrates that 7-membered lactams and large steric hindrances can weaken N-S bond, which promote the reaction. Overall, I feel that this is an excellent work, thus would like to support its publication in Nature Communications. The authors may wish to consider the following minor points.

1. In support information, page S3, entries 6-7. The reaction of ligand is extremely harsh. When triphenylphosphine derivatives were used as ligands, the yield of the product was only 2-3%. Therefore, readers may be interested in the effects of more electron-rich ligands and Buchwald ligands, such as P(Cy)₃, P(tBu)₃HBF₄ and X-Phos.
2. Heck reaction is very valuable as the terminal process of Catellani reaction, but the scope of olefin substrates is relatively narrow. Can styrene derivatives be used as substrates for termination process?
3. Please unify the all template of MestReNova in support information (NMR Spectrum). For example, the chemical shifts on page S41 are black and they are blue on page S52.
4. The author is inspired by the carbonyl group in the leaving group (benzoic acid) of the ortho-amination reagent. ACS Catal. 2018, 8, 12, 11827-11833 describes the coordination mode of leaving group and palladium after ortho-amination and it should be cited.

Reviewer #2 (Remarks to the Author):

Dong and co-workers described a novel Pd/norbornene-catalyzed ortho-C-H thiolation of aryl and heteroaryl iodides for the facile preparation of polysubstituted aromatic sulfur compounds. This transformation was enabled by a new class of sulfenamide electrophiles derived from seven-membered lactams, which was designed based on inspiration from their previous research on ortho-C-H aminations. Interestingly, it's found that the bulky substituent around the nitrogen of the sulfenamides can significantly enhance the thiolation efficiency, which was owing to the elongated N-S bond as revealed by X-ray crystallographic analysis. This ortho-C-H thiolation is compatible with

various ipso functionalization, including Heck, Suzuki or Sonogashira termination. This scalable procedure has broad substrate scope and high chemoselectivity, which make it attractive for synthesis of diversified sulfur-containing aromatic compounds, including aryl sulfoxides and sulfones (via a following selective oxidation). Thus, this work represents important progress in the development of new Catellani-type reactions. The SI is of good quality for the characterization of new compounds.

The reviewer has also noticed that a related work by Gu was recently published in Org Lett. (DOI: 10.1021/acs.orglett.9b00923). However, this doesn't dilute the novelty of this work considering the completely different thiolation reagent used and the much broader scope of aryl iodides and termination methods. Therefore, I recommend the publication of the present manuscript in Nature Communications after addressing the following minor points.

1. The aforementioned manuscript by Gu should be cited and discussed in the main text.
2. The authors may think about recycling the cyclic amide product to regenerate the thiolation reagent, which will be attractive especially for large scale operation.
3. The gram-scale preparation procedure of 5n should be included in SI
4. The quality of NMR spectra of compounds 4i-j and 4l should be improved.

Reviewer #3 (Remarks to the Author):

Recommendation: Publish after minor revisions noted.

Comments:

Dong, Dong and co-workers discovered a new type of electrophiles (t-butyl-substituted sulfenamide) to realize ortho thiolation of aryl and heteroaryl iodides in Catellani reaction. The substituent effect was observed in tuning the sulfonamide reactivity: bulky substituent around the nitrogen can significantly enhance the electrophilicity of the sulfenamides. This manuscript presents a useful strategy to synthesize sulfur-containing aromatic compounds from simple aryl iodides and sulfonamides. The products were well characterized. I would like to suggest this manuscript publish in Nature Communication after minor changes.

- 1) Did the authors try to use other more bulky substituents on sulfonamide except ipr and tBu?
- 2) It is interesting to note that, ortho-unsubstituted aryl iodides must be used the current methodology. Thus, how about using ortho-unsubstituted aryl iodide in the reaction?

3) Did the authors try to use 5 mol% of palladium sources? I would suggest the authors give this information in SI, even it gave low conversion.

4) Some important references which using new types of electrophilic reagents in the Catellani-type reaction should be cited, such as *Org. Lett.* 2019, 21, 3204–3209; *Org. Lett.* 2018, 20, 6640–6645. Furthermore, the important review of Catellani reaction should be cited (DOI: 10.1021/acs.chemrev.9b00079) .

The suggestions and comments from the three reviewers have been well received and appreciated. Here are our responses to these constructive comments:

1. Re: referee 1

- (a) Original comments:** *“In support information, page S3, entries 6-7. The reaction of ligand is extremely harsh. When triphenylphosphine derivatives were used as ligands, the yield of the product was only 2-3%. Therefore, readers may be interested in the effects of more electron-rich ligands and Buchwald ligands, such as P(Cy)₃, P(tBu)₃HBF₄ and X-Phos.”*

This is a great suggestion. In the revised SI (Supplementary Table 1), the results of using more electron-rich ligands, such as P(Cy)₃, P(tBu)₃HBF₄ and XPhos, have been added into the table of the control experiments.

- (b) Original comments:** *“Heck reaction is very valuable as the terminal process of Catellani reaction, but the scope of olefin substrates is relatively narrow. Can styrene derivatives be used as substrates for termination process?”*

We appreciate this excellent suggestion. The use of styrene as the termination reagent has been attempted. Unfortunately, the reaction only gave a complex mixture with trace product. In the revised SI, we added a Supplementary Figure (9) of “Less successful and Unsuccessful Substrates” to clarify the limitation of the reaction scope.

- (c) Original comments:** *“Please unify the all template of MestReNova in support information (NMR Spectrum). For example, the chemical shifts on page S41 are black and they are blue on page S52.”*

We also appreciate this suggestion very much. The template of the NMR spectra has been unified in the revised SI.

- (d) Original comments:** *“the author is inspired by the carbonyl group in the leaving group (benzoic acid) of the ortho-amination reagent. ACS Catal. 2018, 8, 12, 11827-11833 describes the coordination mode of leaving group and palladium after ortho-amination and it should be cited.”*

We fully agree with the reviewer’s suggestion. The aforementioned paper has been cited as ref 30.

2. Re: referee 2

- (a) **Original comments:** *“The aforementioned manuscript by Gu should be cited and discussed in the main text.”*

We fully agree with this suggestion. In the revised manuscript, we added a comment “during the review process of this work, an interesting ortho thiolation using thiosulfonate reagents was reported by Gu” and cited this work as ref 52.

- (b) **Original comments:** *“The authors may think about recycling the cyclic amide product to regenerate the thiolation reagent, which will be attractive especially for large scale operation.”*

This is a great suggestion! The cyclic lactam can be recycled in 86% yield under the standard conditions. This experiment has been added in the revised manuscript (Fig. 3a).

- (c) **Original comments:** *“The gram-scale preparation procedure of 5n should be included in SI”*

We appreciate this suggestion as well. The gram-scale procedure has been added in the revised SI.

- (d) **Original comments:** *“The quality of NMR spectra of compounds 4i-j and 4l should be improved.”*

We are also grateful for this comment. Purification of compounds 4i, 4j and 4l was difficult due to the low polarity of these compounds. To obtain clean NMR spectra, more polar methyl acrylate was used as the terminating reagents for these substrates. The new data and NMR spectra have been added in the revised Table 2 and revised SI.

3. Re: referee 3

- (a) **Original comments:** *“Did the authors try to use other more bulky substituents on sulfonamide except ipr and tBu?”*

This is an excellent suggestion. Besides i-Pr and t-Bu, the methyl substituted sulfenamide has been prepared and tested in the ortho

thiolation reaction. The desired *ortho* thiolation product was obtained in 48% yield, which is better than non-substituted S13 (40%), but worse than *i*-Pr (50%) and *t*-Bu (74%)-substituted sulfenamides. This result has been added to the revised Table 1.

(b) Original comments: *“It is interesting to note that, ortho-unsubstituted aryl iodides must be used the current methodology. Thus, how about using ortho-unsubstituted aryl iodide in the reaction?”*

When *ortho* unsubstituted aryl iodides (i.e. simple iodobenzene) were used as substrates, only 22% of the bis-*ortho*-thiolation product was isolated. It is likely that the chelation of the first phenylthio group makes the second C-H activation step more difficult, which might further cause the catalyst death. This result has been added to the revised SI in the Supplementary Figure 9 of “Less successful and unsuccessful substrates”.

(c) Original comments: *“Did the authors try to use 5 mol% of palladium sources? I would suggest the authors give this information in SI, even it gave low conversion.”*

We appreciate the kind suggestion and performed the suggested study. The result of using 5 mol% Pd has been added in the revised SI.

(d) Original comments: *“Some important references which using new types of electrophilic reagents in the Catellani-type reaction should be cited, such as Org. Lett. 2019, 21, 3204–3209; Org. Lett. 2018, 20, 6640–6645. Furthermore, the important review of Catellani reaction should be cited (DOI: 10.1021/acs.chemrev.9b00079).”*

We are also grateful for these suggestions. These references have been added in the revised manuscript.

REVIEWERS' COMMENTS:

Reviewer #1 (Remarks to the Author):

All issues which the reviewers mentioned have been addressed. I am pleased to recommend this article for publication in its current form.

Reviewer #3 (Remarks to the Author):

The authors have considered all the recommendations that I suggested and made the pertinent amendments. I now support the acceptance of the manuscript in its current form.